# Identification of an LGP2-associated MDA5 agonist in picornavirus-infected cells

**Safia Deddouche[1], Delphine Goubau[1], Jan Rehwinkel[1†], Probir Chakravarty[2], Sharmin Begum[3], Pierre V Maillard[1], Annabel Borg[4], Nik Matthews[3], Qian Feng[5], Frank J M van Kuppeveld[5], Caetano Reis e Sousa[1]***

[1]Immunobiology Laboratory, Cancer Research UK, London Research Institute, London, United Kingdom; [2]Bioinformatics Laboratory, Cancer Research UK, London Research Institute, London, United Kingdom; [3]Clonal Sequencing Laboratory, Cancer Research UK, London Research Institute, London, United Kingdom; [4]Protein Purification Laboratory, Cancer Research UK, London Research Institute, London, United Kingdom; [5]Virology Division, Department of Infectious Diseases and Immunology, Utrecht University, Utrecht, Netherlands

**\*For correspondence:** caetano@cancer.org.uk

**Present address:** [†]Medical Research Council Human Immunology Unit, Radcliffe Department of Medicine, Medical Research Council Weatherall Institute of Molecular Medicine, University of Oxford, Oxford, United Kingdom

**Competing interests:** The authors declare that no competing interests exist.

**Reviewing editor**: Zhijian J Chen, Howard Hughes Medical Institute, University of Texas Southwestern Medical School, United States

**Abstract** The RIG-I-like receptors RIG-I, LGP2, and MDA5 initiate an antiviral response that includes production of type I interferons (IFNs). The nature of the RNAs that trigger MDA5 activation in infected cells remains unclear. Here, we purify and characterise LGP2/RNA complexes from cells infected with encephalomyocarditis virus (EMCV), a picornavirus detected by MDA5 and LGP2 but not RIG-I. We show that those complexes contain RNA that is highly enriched for MDA5-stimulatory activity and for a specific sequence corresponding to the L region of the EMCV antisense RNA. Synthesis of this sequence by in vitro transcription is sufficient to generate an MDA5 stimulatory RNA. Conversely, genomic deletion of the L region in EMCV generates viruses that are less potent at stimulating MDA5-dependent IFN production. Thus, the L region antisense RNA of EMCV is a key determinant of innate immunity to the virus and represents an RNA that activates MDA5 in virally-infected cells.

## Introduction

Viral infection in mammals triggers a rapid innate immune response involving the production of antiviral proteins and proinflammatory mediators, prominent among which are the type I interferons (IFN-α/β; hereafter, IFN) (*Stetson and Medzhitov, 2006*; *Takaoka and Yanai, 2006*). IFNs are secreted cytokines that act on all nucleated cells to induce the transcription of more than 300 IFN-stimulated genes, whose products collectively limit virus replication and spread (*Haller et al., 2007*; *Schoggins and Rice, 2011*). IFN gene transcription is triggered by the activation of pattern recognition receptors that detect viral invasion. These receptors include members of the RIG-I-like receptor (RLR) family (*Goubau et al., 2013*), a sub-group of DExD/H-box helicases that surveys the cytosol for the presence of atypical RNAs associated with viral infection. The RLR family comprises three members: RIG-I (retinoic acid-induced gene I), MDA5 (melanoma differentiation-associated gene 5), and LGP2 (laboratory of genetics and physiology 2). Binding of agonistic RNA by RIG-I or MDA5 triggers a signalling cascade that leads to the activation of transcription factors, including several of the IFN regulatory factors (IRFs), which translocate into the nucleus to induce expression of IFN and other genes. LGP2 lacks a signalling domain and cannot act in the same manner. Rather, LGP2 is thought to potentiate MDA5-dependent IFN induction although the exact mechanism by which this occurs remains unclear (*Satoh et al., 2010*; *Bruns et al., 2012*; *Childs et al., 2013*).

**eLife digest** A virus is basically molecules of DNA or RNA inside a protein shell, and in order to reproduce, it must infect a living cell and take control of it. However, the attacked cell will fight back and try to eliminate the invader. Activation of this so-called innate immune response requires the host cells to recognize that they have been infected, which they do by detecting the tell-tale molecules that indicate the presence of the virus.

When an RNA virus infects a cell, the tell-tale molecules are often atypical RNA molecules carried by the virus or produced as the virus replicates. Recognition of this 'foreign' material by receptor proteins inside the cell triggers the production of molecules called interferons, which activate the innate defence systems that eliminate the virus.

Different receptor proteins recognize different RNA viruses. For example, a receptor called MDA5 is known to respond to the picornaviruses, some of which can cause inflammation of the brain and heart muscle. However, the identities of the specific RNA molecules that are recognized by the MDA5 receptor have not been known. Deddouche et al. have now identified one such RNA molecule with the help of a second receptor protein, called LGP2.

The LGP2 receptor is not able to give the signal to produce interferons, so it is thought to work by binding to the RNA molecule to form a complex that is then relayed to MDA5 to give this signal. By isolating the complexes of LGP2 receptor from picornavirus-infected cells and sequencing the associated RNA, it was possible to identify the mystery RNA trigger. Deddouche et al. then tested picornaviruses in which this piece of RNA had been deleted from the genome, and found that the mutant viruses triggered a much weaker interferon response. By providing insight into the ways that some viruses are detected by the innate immune system, this research may inform future work on the development of new treatments to control viral infection.

---

MDA5 and RIG-I are activated by different RNA viruses (*Kato et al., 2006*). RIG-I is non-redundant for detection of influenza or Sendai virus but dispensable for responses to picornaviruses whereas the opposite is the case for MDA5 (*Kato et al., 2006*; *2008*). The basis for these differences stems from the ability of RIG-I and MDA5 to recognize different RNA patterns. RIG-I binds to and is activated by base-paired RNA containing a 5′ triphosphate extremity (*Hornung et al., 2006*; *Pichlmair et al., 2006*; *Schlee et al., 2009*; *Schmidt et al., 2009*) as found in influenza and Sendai virus genomes (*Baum et al., 2010*; *Rehwinkel et al., 2010*; *Weber et al., 2013*). In contrast, MDA5 recognizes RNA independently of 5′ phosphates and can be activated in cells by transfection of synthetic poly I:C or the double-stranded (ds) RNA genome segments of reovirus (*Kato et al., 2006*, *2008*). Based on such observations, it has been concluded that MDA5 detects long dsRNA generated during virus infection. However, natural MDA5 agonists derived from infected cells remain poorly characterised. Some studies have suggested that they might correspond to high molecular weight RNA complexes containing double and single-stranded (ss) regions (*Pichlmair et al., 2009*). Other reports have proposed that relevant MDA5 agonists are specific viral replication intermediates (*Feng et al., 2012*; *Triantafilou et al., 2012*). Finally, it has been suggested that MDA5 might also be activated by RNA products of RNAseL cleavage (*Malathi et al., 2010*; *Luthra et al., 2011*) or by mRNA bearing unmethylated cap structures (*Züst et al., 2011*). Altogether, these studies provide a glimpse into the possible nature of MDA5 agonists in virally-infected cells but fall short of identifying them with precision.

Immunoprecipitation of RIG-I from cells infected with influenza A virus (IAV) or Sendai virus has allowed the identification of physiological RIG-I agonists (*Baum et al., 2010*; *Rehwinkel et al., 2010*). In this study, we use an analogous approach to investigate the nature of MDA5 agonists in cells infected with encephalomyocarditis virus (EMCV), a member of the *Cardiovirus* genus of picornaviruses. Picornaviruses are single-stranded positive-strand (sense) RNA viruses that replicate in infected cells via a negative-strand (antisense) intermediate. We purify RNA directly from complexes obtained by immunoprecipitation of LGP2 and show that this method enriches for MDA5 stimulatory RNA corresponding to a portion of the EMCV antisense RNA. Deletion of the region encoding this antisense RNA generates viruses that produce less stimulatory RNA and are less potent at inducing IFN in infected cells or mice. Conversely, in vitro synthesis of the same sequence generates an MDA5 agonistic RNA. Thus, a discrete region of the EMCV negative-strand RNA acts as a physiologically-relevant MDA5 agonist in infected cells.

## Results

### EMCV replication is required for MDA5/LGP2-dependent IFN induction

To confirm that both MDA5 and LGP2 are required for IFN responses to EMCV (*Kato et al., 2006*; *Satoh et al., 2010*), we used mouse embryonic fibroblasts (MEFs) carrying null mutant alleles of the genes *Ifih1* and *Dhx58* encoding MDA5 and LGP2, respectively. We infected *Ifih1⁻/⁻* (MDA5-deficient), *Ifih1⁻/⁺* (MDA5-sufficient), *Dhx58⁻/⁻* (LGP2-deficient) or *Dhx58⁺/⁺* (LGP2-sufficient) MEFs and assessed the induction of IFN-β and the interferon-stimulated protein IFIT-1. The upregulation of *Ifit1* or *Ifnb1* mRNA was greatly impaired in MDA5- or LGP2-deficient MEFs infected with EMCV (*Figure1—figure supplement 1A,B*). The same cells responded normally to RIG-I-dependent viruses such as IAV and to known RIG-I agonists such as in vitro transcribed (IVT) RNA (*Figure1—figure supplement 1A,B*). To begin to define the MDA5/LGP2 agonist, we isolated the EMCV genome from purified EMCV particles and transfected it into reporter cells together with a plasmid encoding a luciferase gene under the control of the IFN-β promoter. As reporter cells, we used an easily transfectable subclone of HEK293 cells that expresses all RLRs (*Figure 1—figure supplement 2A*) and can respond, albeit weakly, to MDA5 agonists (data not shown; *Figure 1*). Because transfection of positive-stranded viral RNA can lead to viral replication (even though EMCV replicates in HEK293 cells only poorly), we performed the IFN reporter assay in the presence of ribavirin, an inhibitor of viral RNA synthesis. As seen in *Figure 1A*, EMCV genomes did not stimulate the IFN-β reporter, in contrast to the genomes of IAV, which directly activate RIG-I (*Baum et al., 2010*; *Rehwinkel et al., 2010*; *Weber et al., 2013*). To determine whether viral replication generates stimulatory RNA, we extracted total RNA from HeLa cells that had been infected with EMCV in the presence or absence of ribavirin. RNA isolated from cells in which EMCV viral replication had been permitted to take its course (DMSO control) potently induced the IFN-β reporter upon transfection into HEK293 cells (*Figure 1B*). In contrast, RNA extracted from HeLa cells treated with ribavirin was non-stimulatory (*Figure 1B*). Treatment of the reporter HEK293 cells themselves with ribavirin did not affect the response (*Figure 1—figure supplement 2B,C*), which indicates that the stimulatory RNA is preformed in EMCV-infected HeLa cells. Furthermore, the response in the HEK293 reporter cells was dependent on MDA5 as demonstrated using RNA interference-mediated MDA5 knockdown (*Figure 1—figure supplement 2D*). Altogether these data indicate that MDA5 and LGP2 activation results exclusively from RNA generated during active EMCV replication, as recently suggested (*Feng et al., 2012*; *Triantafilou et al., 2012*).

One feature of the replication cycle of positive-strand RNA viruses is the generation of a negative-strand RNA that, together with the annealed positive strand, forms a long dsRNA structure. To characterise the 'strandedness' of the IFN stimulatory RNA generated upon EMCV replication, we extracted total RNA from non-infected or either IAV or EMCV-infected HeLa cells and separated it into ds and ssRNA fractions (*Feng et al., 2012*). As expected (*Pichlmair et al., 2006*), the ssRNA but not the long dsRNA fraction extracted from cells infected with IAV, stimulated the IFN reporter (*Figure 1C,D*). In contrast, both ds and ssRNA fractions from cells infected with EMCV triggered the IFN-β reporter (*Figure 1C,D*). Similarly, treatment of total RNA from EMCV-infected cells with RNase A or RNase III, which digest unpaired or base-paired RNA, respectively, strongly reduced stimulatory activity (*Figure 1E*). In contrast, removal of 5' phosphates by digestion with calf intestinal phosphatase (CIP) or 5' polyphophatase (PP) did not impact stimulatory activity although it abrogated that of IVT control RNA (*Figure 1E*). Altogether these data indicate that MDA5/LGP2 stimulatory RNA accumulates in EMCV-infected cells during virus replication and contains both ssRNA and dsRNA features, as previously suggested (*Pichlmair et al., 2009*).

### Enrichment of MDA5 agonists from EMCV-infected cells by LGP2 pulldown

To further purify the RNA responsible for activation, we established a method to isolate LGP2-associated RNA from infected cells. We immunoprecipitated (IP) LGP2 from EMCV-infected HeLa cells transiently expressing a FLAG-tagged LGP2 protein (*Figure 2A*), extracted RNA from the precipitates and analysed its stimulatory activity in reporter cells (*Figure 2B*). Notably, RNA associated with the LGP2 precipitates, but not with control (ctrl) precipitates, stimulated the IFN-β luciferase reporter in HEK293 cells and induced expression of the IFNB1 gene in HeLa cells (*Figure 2B,C*). Moreover, when compared to input material, stimulatory activity was significantly enriched in the LGP2-immunoprecipitate and was selectively depleted from the unbound fraction (*Figure 2B,C*). We additionally transfected the LGP2-associated RNA into MEFs and confirmed its ability to stimulate IFN production by ELISA (*Figure 2D*).

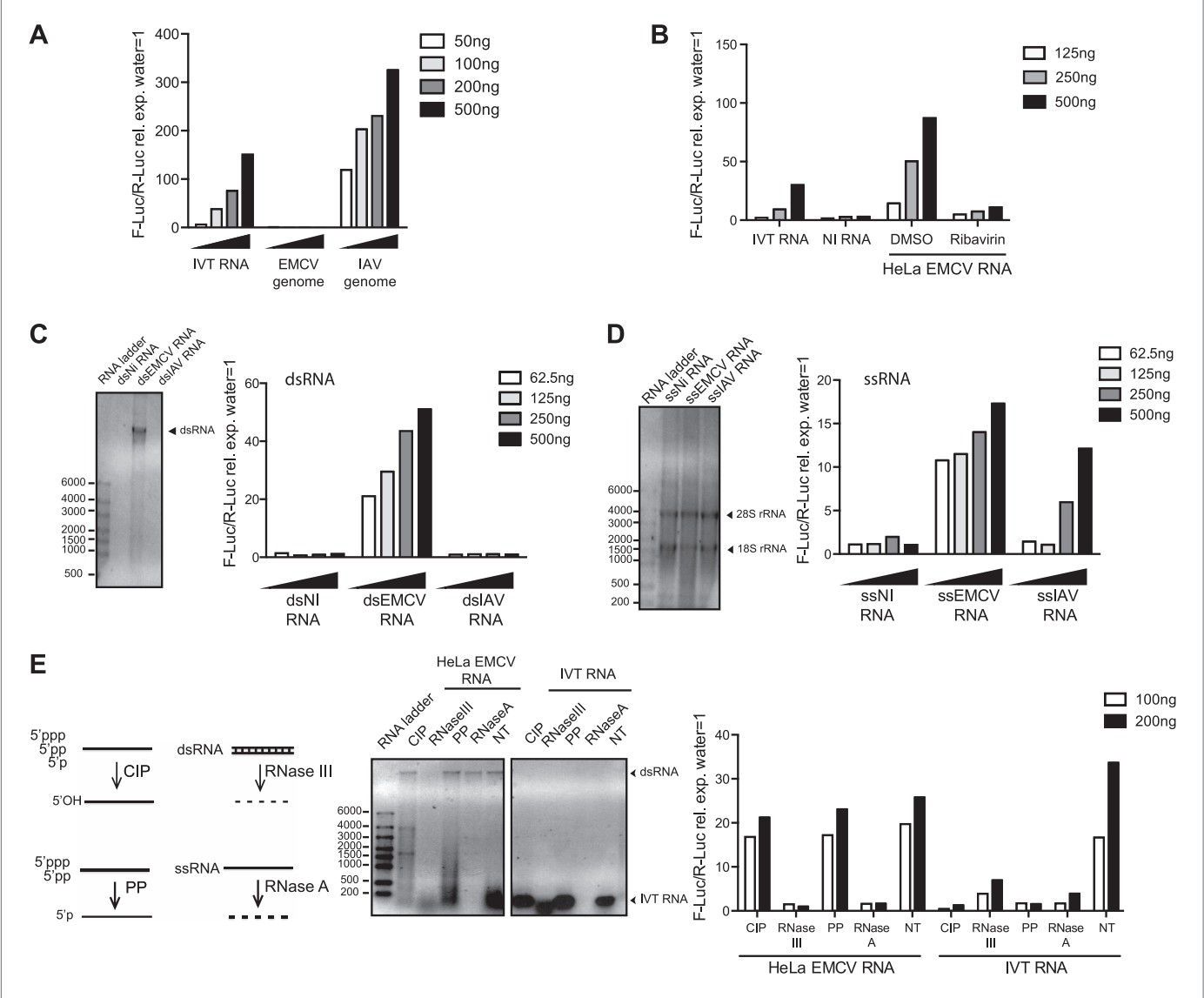

**Figure 1**. IFN-α/β induction requires EMCV replication. (**A**) EMCV and IAV RNA genomes were extracted from purified viral particles and tested at the indicated doses in an IFN-β promoter luciferase reporter assay in HEK293 cells in the presence of ribavirin. IVT-RNA was included as a positive control. (**B**) HeLa cells were infected with EMCV (MOI 1) for 16 hr in the presence of DMSO or ribavirin. RNA was extracted (HeLa EMCV RNA) and tested at the indicated doses in the IFN-β promoter luciferase reporter assay in HEK293 cells. RNA extracted from uninfected HeLa cells (NI RNA) and IVT-RNA were included as negative and positive controls, respectively. (**C** and **D**) HeLa cells were either not infected (NI) or infected with EMCV or IAV at MOI 1 for 16 hr. RNA was extracted from cell lysates and separated into double-stranded (ds; panel **C**) or single-stranded (ss; panel **D**) fractions. The fractions were analysed on a 1% agarose gel and the indicated amounts of RNA were then tested at the indicated doses in the IFN-β promoter luciferase reporter assay in HEK293 cells. dsRNA bands are indicated on the gel picture. ssRNA runs as a smear and only the ribosomal RNA bands are identifiable, as indicated. ssRNA was used as a ladder. Although the experiment depicted was carried out in the absence of ribavirin, identical results were obtained in the presence of the drug indicating that the stimulatory capacity of ssRNA fractions is not due to subsequent replication and formation of dsRNA (data not shown). (**E**) Total RNA from HeLa cells infected with EMCV at MOI 1 for 16 hr (HeLa EMCV RNA) or control IVT RNA was either left untreated (NT) or digested with calf intestinal phosphatase (CIP), base-paired specific RNase (RNaseIII), 5' RNA polyphosphatase (PP), or single-strand-specific RNAse (RNaseA). Digested RNA samples were analysed on a 1% agarose gel and the indicated amounts of RNA were tested in the IFN-β promoter luciferase reporter assay in HEK293 cells. One representative of three (**A**, **C**, **D**) or two (**B** and **E**) experiments is shown.

The following figure supplements are available for figure 1:

**Figure supplement 1**. LGP2 and MDA5 are required for IFN-α/β production in response to EMCV.

**Figure supplement 2**. Ribavirin abrogates EMCV RNA infectivity but does not decrease IFN-β reporter activity in 293 cells.

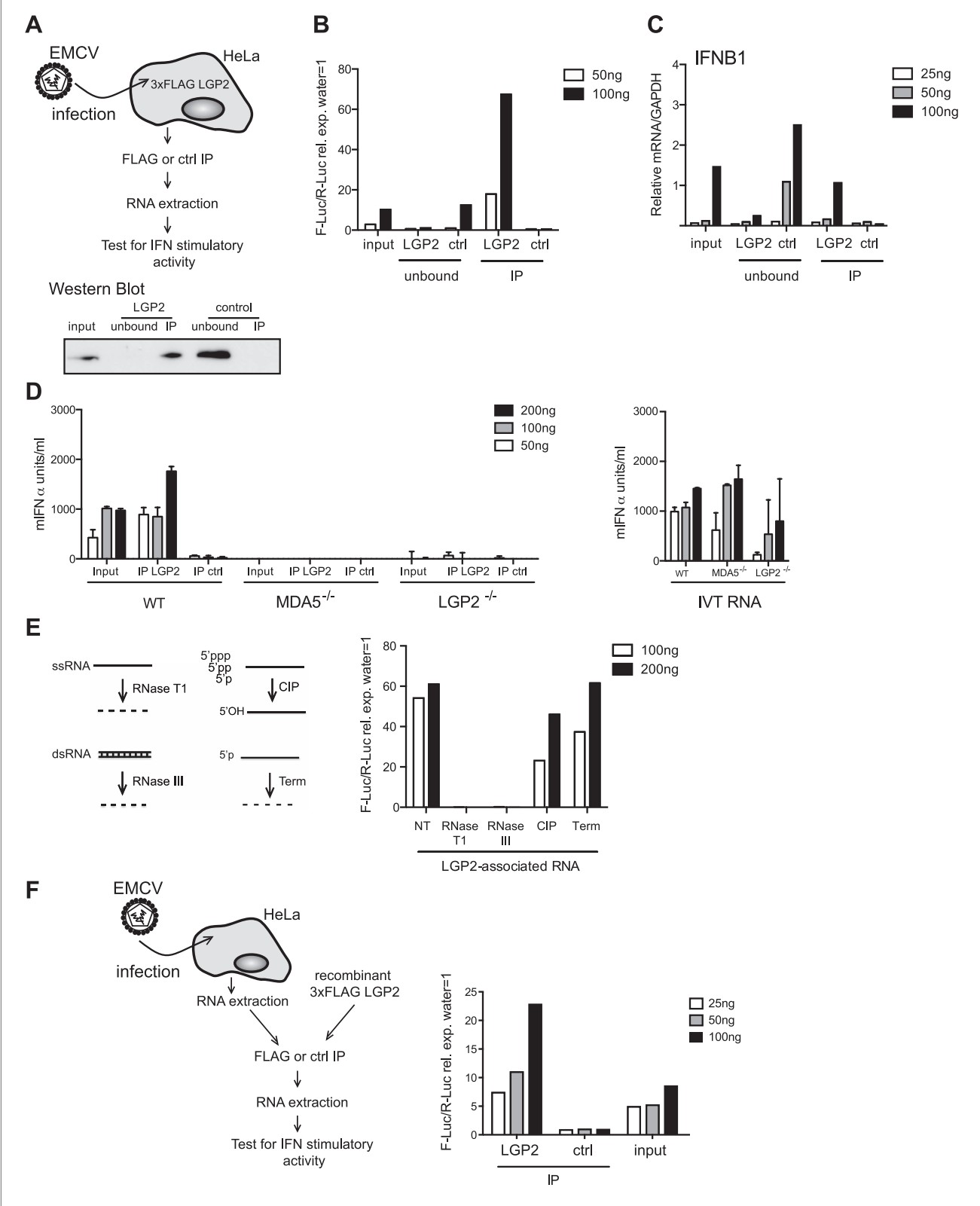

**Figure 2**. LGP2 pulldown captures MDA5 agonistic RNA from EMCV-infected cells. (**A**) Schematic representation of the experimental setup for LGP2 immunoprecipitation (IP). Precipitation efficiency was routinely verified by immunoblotting with an anti-FLAG antibody; an example is shown in the lower panel. (**B** and **C**) The indicated amounts of RNA from EMCV-infected FLAG-LGP2-expressing HeLa cells (input), RNA associated with LGP2 or control

*Figure 2. Continued on next page*

*Figure 2. Continued*

(ctrl) immunoprecipitations (IP), or RNA remaining after LGP2 or control precipitations (unbound) were tested for the ability to stimulate the IFN-β promoter reporter assay in HEK293 cells (**B**) or induce IFNB1 mRNA (normalised to GAPDH) in HeLa cells (**C**). (**D**) The indicated amounts of RNA from samples processed as in (**B**) and (**C**) were transfected into WT, LGP2-deficient, or MDA5-deficient MEFs. Supernatants were harvested 16 hr later and mIFN-α levels measured by ELISA (left panel). IVT RNA transfections were used as positive controls (right panel). Error bars represent the standard deviation of three replicate transfections. (**E**) LGP2-associated RNA was not treated (NT) or digested with RNaseT1 (specific for ssRNA), RNaseIII (specific for base-paired RNA), CIP or Terminator (Term, an RNase specific for 5'monophosphate RNA) and subsequently tested for the ability to stimulate the IFN-β reporter in HEK293 cells. The activity of all enzymes was validated in control samples (not shown). (**F**) RNA from HeLa cells infected with EMCV (input) was incubated with recombinant FLAG-tagged LGP2 protein and anti-FLAG or control IP was performed as in (**A**). IP-associated RNA was isolated and tested, in parallel with input RNA, at the indicated doses in the IFN-β promoter luciferase reporter assay on HEK293 cells. Schematic representation of the experiment is shown on the left, results are presented on the right. One representative of the three (**A**–**C**) or two (**D**–**F**) experiments is shown.

Importantly, activity was lost in LGP2- or MDA5-deficient MEFs (*Figure 2D*), indicating that the LGP2-associated RNA isolated from immunoprecipitates is a pure MDA5/LGP2 agonist. Consistent with this notion, its IFN stimulatory capacity was unaffected by treatment with CIP, which inactivates most RIG-I agonists, or Terminator (Term), which digests RNA with 5′ monophosphates (*Figure 2E*). Furthermore, digestion with RNaseT1 and RNaseIII completely abolished stimulatory potential (*Figure 2E*), again suggesting the presence of unpaired and base-paired RNA regions as previously observed with total RNA from EMCV-infected cells (*Figure 1E*). Finally, we incubated purified RNA from EMCV-infected HeLa cells together with recombinant FLAG-tagged LGP2 (*Figure 2F*). FLAG immunoprecipitation allowed enrichment for stimulatory activity when compared to a control IP, demonstrating that stimulatory RNA can associate with LGP2 in vitro in the absence of additional proteins (*Figure 2F*). Altogether these results indicate that LGP2 selectively associates with a pool of MDA5 agonists in EMCV-infected cells and that LGP2 IP is a suitable approach to enrich for such agonists.

## L region antisense EMCV RNA is selectively enriched in LGP2 precipitates

Having established a method to purify IFN stimulatory RNA from LGP2/RNA complexes isolated from EMCV-infected HeLa cells, we subjected it to deep sequencing analysis. We pooled RNA extracted from multiple independent control or LGP2 IPs and carried out Illumina sequencing in duplicate ('Material and methods'). Approximately, 30 million reads of 60 nts in length were obtained from each sequencing sample and were mapped to the human and the EMCV genomes. The total number of reads mapping to EMCV represented around 30% in LGP2 IP samples vs only 4% and 6% in ctrl IP and input, respectively (*Tables 1 and 2*; *Figure 3A*), which indicated that the RNA in the LGP2 IP is specifically enriched for EMCV-derived sequences. To allow better comparison across samples, the number of reads from LGP2 IP, ctrl IP and input samples was first normalised to the total number of reads (displayed as reads/million) and then aligned to the EMCV genome (*Figure 3*, *Figure 3—figure supplement 1*). Surprisingly, the distribution of sequences in the LGP2 IP sample was not uniform but displayed a number of discrete peaks concentrated in the 5′ region of the EMCV genome. In particular, one peak from position 735 nts to 905 nts on the antisense RNA was strongly enriched (25,000 reads/million) over input and ctrl IP samples (non detectable and 60 reads/million, respectively) (*Figure 3B*). A smaller peak (±5,000 reads/million) in the corresponding part of the sense strand was also enriched in LGP2 IP samples (*Figure 3B*). This region encodes the leader (L) protein of EMCV and is henceforth referred to as the L region.

**Table 1.** Total number of reads aligning to the EMCV genome in LGP2 IP, ctrl IP, or input samples

|                  | LGP2 IP    | ctrl IP    | input      |
|------------------|------------|------------|------------|
| Number of reads* | 31,662,255 | 37,235,332 | 35,725,972 |
| EMCV             | 2,747,427  | 243,472    | 1,380,128  |
| EMCV (+)         | 1,305,129  | 239,625    | 1,380,058  |
| EMCV (−)         | 1,442,298  | 3,847      | 70         |

*Total numbers of reads, reads matching both strands (EMCV), sense strand (EMCV (+)) or antisense strand (EMCV (−)) of EMCV RNA sequences.

**Table 2.** Percentage of reads mapping either the sense (+) or the antisense (−) strand in the L region compared to the full length EMCV genome

| | L region | | Total EMCV | |
|---|---|---|---|---|
| | (+) | (−) | (+) | (−) |
| LGP2 IP | 19.49 | 80.51 | 47.50 | 52.50 |
| ctrl IP | 78.55 | 21.45 | 98.17 | 1.83 |
| input | 100.00 | 0.00 | 100.00 | 0.00 |

To validate these findings, we used strand-specific RT-PCR with primers for the L antisense or, as a control, the VP1 antisense regions (primer localisation indicated by red arrows on the schematic representation of the EMCV genome in *Figure 3B*). This analysis confirmed that the L antisense but not the VP1 antisense region, was enriched in LGP2 IP samples compared to ctrl IPs and input material (*Figure 3C*). These findings were further validated in the LGP2 in vitro reconstitution assay, which confirmed that RNA that binds to LGP2 is enriched for the L antisense region (*Figure 3—figure supplement 2*). In sum, LGP2 association with EMCV RNA is strongly biased towards a discrete area of the negative (antisense) strand of the L protein-encoding region.

We next examined if the L antisense (AS) RNA sequence was sufficient to trigger an MDA5-dependent IFN response. We generated, by in vitro transcription, RNAs corresponding either to the AS or the sense strand of the EMCV L region. After CIP treatment to remove any RIG-I-stimulatory activity linked to the presence of the 5′ triphosphates, we tested the stimulatory potential of these RNAs in MEFs deficient for the *Ddx58* gene encoding RIG-I or in MDA5-deficient MEFs by IFN-β promoter luciferase assay (*Figure 4A,B*). The RNA containing the L AS derived sequence was clearly stimulatory in contrast to the one derived from the L sense sequence. Moreover, the activity of the CIP-treated L AS was greatly reduced in the MDA5-deficient but not RIG-I-deficient MEFs (*Figure 4A*). As a control, we used non-CIP treated IVT RNA corresponding to the sense sequence of the neomycin gene (*Rehwinkel et al., 2010*), which showed the expected RIG-I dependence (IVT RNA, *Figure 4C*). We conclude that an IVT RNA corresponding to the EMCV L AS RNA sequence found in LGP2 immunoprecipitates (*Figure 3*) can trigger an MDA5-dependent IFN response.

## The L region of EMCV is required for generation of LGP2-associated stimulatory RNA

To ask whether the L region of the EMCV genome is important for the generation of IFN stimulatory RNA, we used mutant EMCV viruses with partial (EMCV ΔL$_{ac}$ and EMCV ΔL$_{zn}$) or complete (EMCVΔL) deletions of that region (*Dvorak et al., 2001*; *Figure 5A*). We infected HeLa cells, extracted RNA and subjected it to RT-PCR analysis. Using specific primers (shown as arrows in *Figure 5A*), we confirmed the loss of the L region in the mutant viruses (*Figure 5B*, left panel). Because all three viruses are attenuated due to the absence of the L protein, a known antagonist of IFN induction (*Hato et al., 2007*), we also amplified the Vp1 region to verify that similar levels of EMCV RNA were present in all samples (*Figure 5B*, right panel). We then assessed the IFN stimulatory potential of these samples using the IFN-β reporter assay. In all cases, RNA extracted from HeLa cells infected with EMCV ΔL$_{ac}$, ΔL$_{zn}$ or ΔL was slightly less stimulatory than RNA extracted from cells infected with wild-type (WT) EMCV (*Figure 5C*). More importantly, only a low amount of stimulatory RNA was recovered from LGP2 IPs following infection with the mutant viruses in contrast to infection with WT EMCV (*Figure 5D*). These results are consistent with the earlier indications that the major species of stimulatory RNA associating with LGP2 in EMCV-infected cells derives from the L antisense region.

Deletion of the L region results in a virus that is both depleted of L RNA and L protein. To verify that the lack of LGP2-associated stimulatory RNA upon infection with ΔL$_{ac}$, ΔL$_{zn}$ or ΔL viruses was due to the absence of L region RNA rather than absence of L protein function, we additionally used an EMCV Zn$_{C19AC22A}$ strain. This virus carries two mutations in the zinc domain of the L protein, which inactivate function (*Hato et al., 2007*) but should not impact on the production of L RNA (even if that RNA now carries two nucleotide substitutions). Reassuringly, the amount of stimulatory RNA in total cell extracts (*Figure 5E*) or associated with LGP2 precipitates (*Figure 5F*) was comparable upon infection with EMCV Zn$_{C19AC22A}$ and EMCV WT even though, as before, it was markedly reduced after EMCVΔL infection

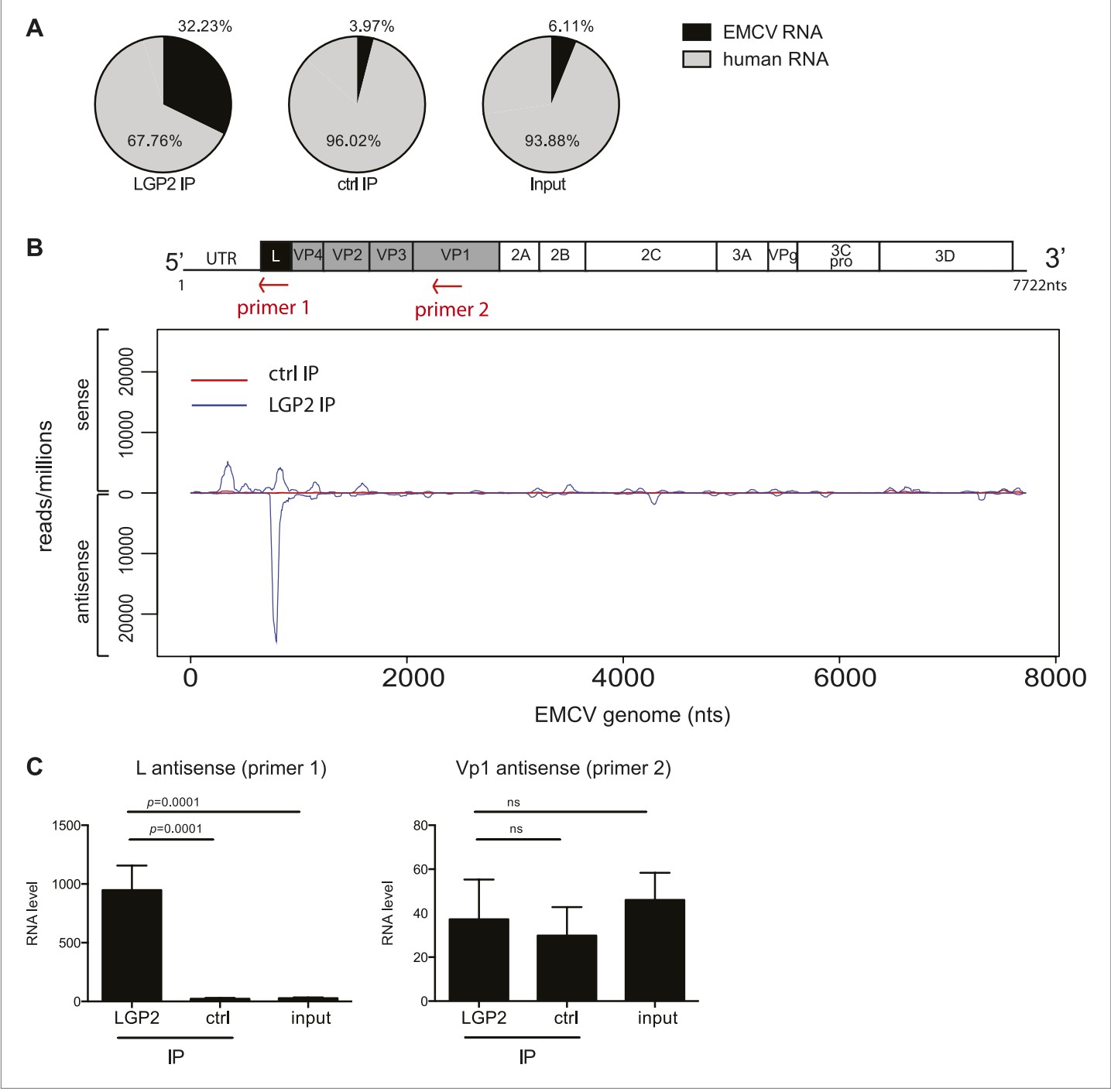

**Figure 3**. The L antisense RNA region is enriched in LGP2 pulldowns from EMCV-infected cells. (**A**) RNA from LGP2 IP, control (ctrl) IP, or total RNA (input) from EMCV-infected cells (*Figure 2A,B*) was sequenced. Reads corresponding to human or EMCV sequences are shown as a percentage of the total number of reads that could be aligned to any sequence in bioinformatic databases. (**B**) All reads obtained from LGP2 IP or ctrl IP from EMCV infected cells were normalised to the number of reads per million to allow for comparison across samples. Results were mapped to the EMCV genome (depicted above; the red arrows indicate the respective position of the primer used for the reverse transcription prior to quantitative RT-PCR in [**C**]). The total numbers of reads are shown in *Table 1*. The vertical axis shows the number of normalised reads mapping to a particular position on the EMCV genome (horizontal axis). The positive and negative numbers represent reads that align to the sense or the antisense strand, respectively. Numbering along the x-axis indicates nucleotide position on the sense (+) strand. One experiment of two is shown. (**C**) The amount of L antisense (AS) (primer 1 in **B**) or VP1 AS (primer 2 in **B**) RNA was analysed by strand-specific RT-PCR in LGP2 IP, ctrl IP, and input samples from an independent experiment. Vertical

*Figure 3. Continued on next page*

*Figure 3. Continued*
axis represents RNA level, calculated relative to the data obtained from a standard curve of cDNA from EMCV-infected cells. Error bars represent the standard deviation of four independent samples. ns = not significant.
The following figure supplements are available for figure 3:

**Figure supplement 1**. Comparison of read-distribution along the EMCV genome for LGP2-associated RNA and input material.

**Figure supplement 2**. LGP2 directly binds L region antisense RNA.

(*Figure 5E,F*). We conclude that the lack of stimulatory LGP2-associated RNA observed after infection with EMCVΔL is specifically due to the loss of L region RNA rather than the loss of L protein function.

## L region RNA is required for innate detection of EMCV

We subsequently assessed the importance of the L RNA sequence for IFN responses in primary cells by comparing infection with EMCVΔL and EMCV $Zn_{C19AC22A}$ mutant viruses. We used dendritic cells grown from mouse bone marrow (BM-DCs), which produce vast amounts of IFNs in response to virus infection (*Diebold et al., 2003*). To allow efficient replication of the L protein-deficient viral strains, BM-DCs were derived from IFN-α/β receptor-deficient mice (IFNAR1 knockout [KO]). To monitor EMCV detection and the activation of the downstream RLRs signalling pathway, we assessed the induction of both Ifit1 and Ifnb1, which are direct transcriptional targets of IRF-3 (*Grandvaux et al., 2002*). Upon infection with WT EMCV, induction of *Ifit1* and *Ifnb1* was limited (*Figure 6A,B*). It was markedly greater in response to infection with EMCV $Zn_{C19AC22A}$, which encodes the non-functional mutant L protein (*Figure 6A,B*), consistent with the fact that the L protein inhibits IFN induction (*Hato et al., 2007*). In contrast, infection with EMCVΔL lacking both L protein and EMCV L region RNA induced lower levels of *Ifnb1* or *Ifit1* than infection with EMCV $Zn_{C19AC22A}$ at two different multiplicities of infection (MOI) (*Figure 6A,B*). EMCV ΔL and EMCV $Zn_{C19AC22A}$ replicated to similar levels, indicating that any differences in stimulation by these viruses are not caused by variations in viral RNA levels (*Figure 6C*). Similar results were obtained when the mutations were introduced into the EMCV mengo strain background (data not shown). These data show that the L region RNA of EMCV is important for RLR stimulation and viral restriction in infected cells and that this is independent on its ability to encode a functional L protein.

In parallel, we also examined the IFN response to infection with an EMCV mengo strain mutant virus carrying the L region of the foot-and-mouth disease virus (FMDV). FMDV belongs to a different picornavirus genus and the FMDV L region sequence and FMDV leader protein share little similarity with those of EMCV. Notably, the induction of both Ifit1 and Ifnb1 in response to EMCV $L_{FMDV}$ was comparable to that induced by EMCVΔL (*Figure 6A,B*) and replication of the two strains was indistinguishable.

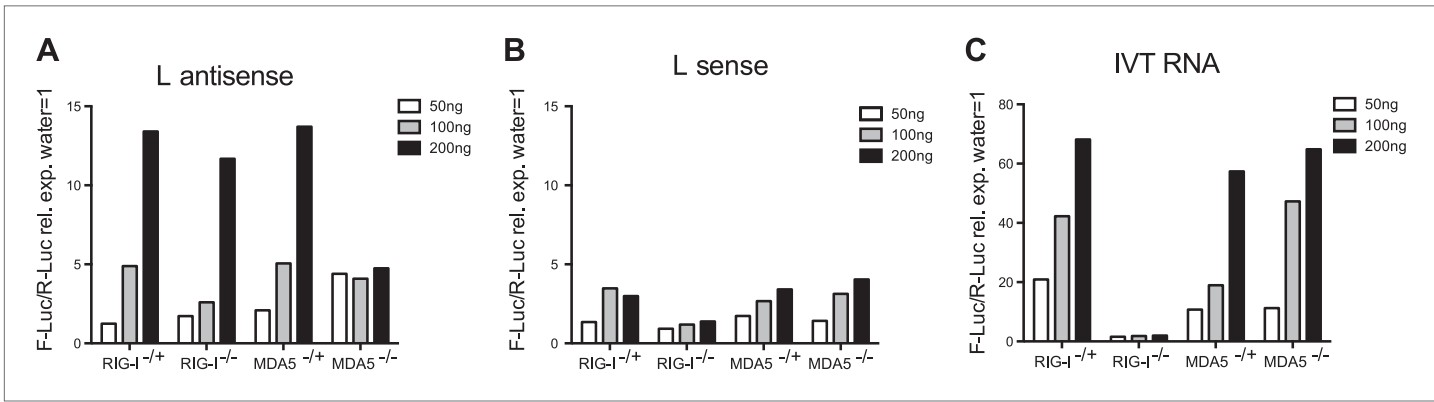

**Figure 4**. In vitro transcribed L AS RNA triggers an MDA5-dependent IFN response. (**A–C**) L antisense (AS) (**A**), L sense (**B**) or IVT RNA (**C**) sequences were in vitro transcribed and all RNA except from the control IVT RNA (**C**) were CIP treated to remove any 5' phosphates. The indicated amount of RNA were then transfected into RIG-I-, MDA5-deficient or sufficient MEFs expressing the IFN-β reporter. Reporter activity was measured 16 hr later. One of two experiments is shown.

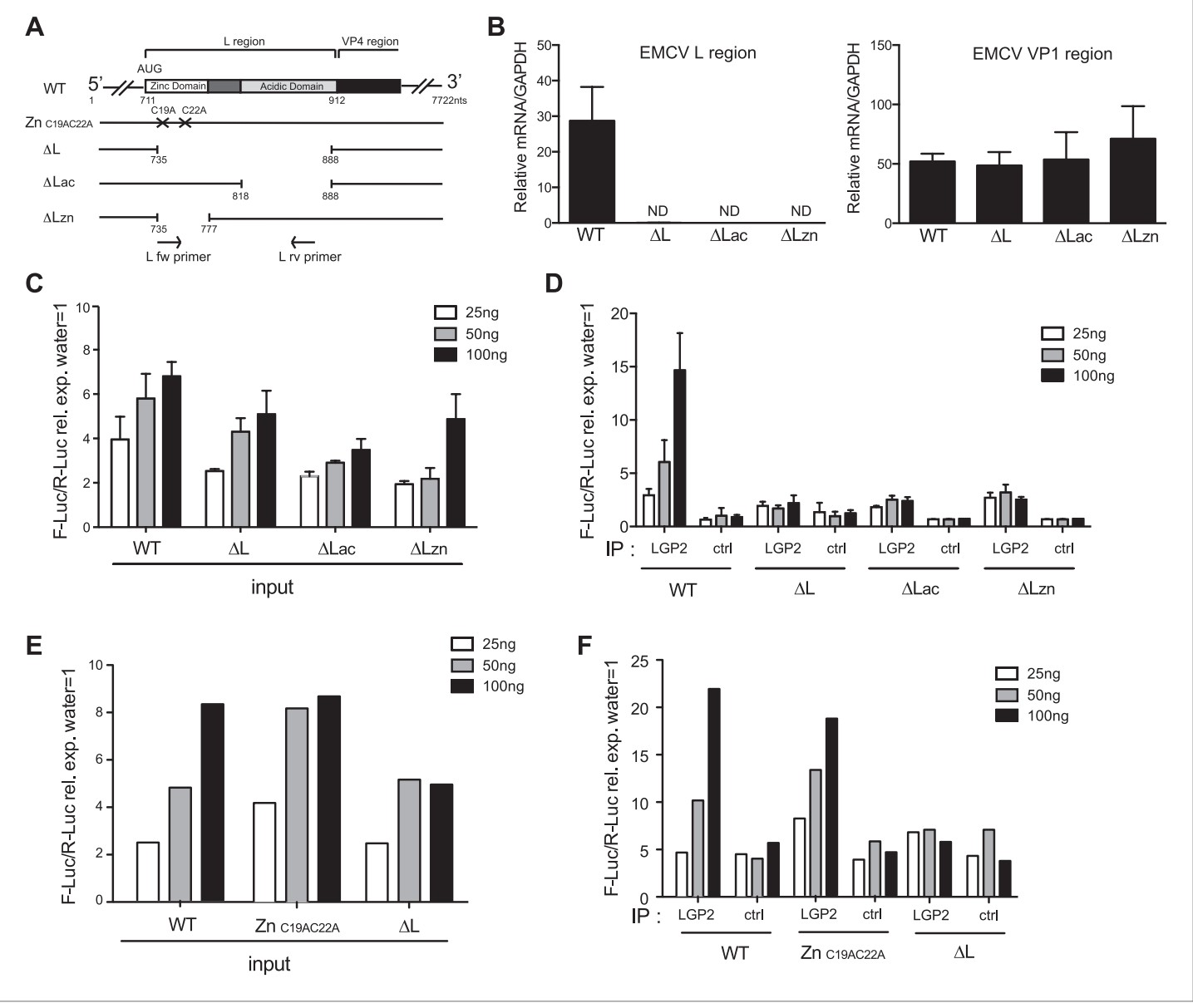

**Figure 5**. The L region of EMCV is required for the generation of LGP2-associated stimulatory RNA. (**A**) Schematic representation of the L region of EMCV genome and L region mutant viruses used in this study. The crosses indicate the position of the two point mutations in EMCV Zn$_{C19AC22A}$. The marked positions indicate the limits of the deletions present in EMCV ΔL, ΔLac, and ΔLzn, respectively. (**B–F**) FLAG-LGP2-expressing HeLa cells were infected with EMCV WT, Zn$_{C19AC22A}$, ΔL, ΔLac, or ΔLzn viruses at MOI 1 for 16 hr before lysis and total RNA extraction (input; **B**, **C**, **E**) or lysis followed by LGP2 or control (ctrl) immunoprecipitation and RNA extraction from precipitates (IP; **D** and **F**). (**B**) Quantitative RT-PCR analysis of viral sequences in input samples using primers specific for the L region (left panel) or the Vp1 region (right panel) and normalised to GAPDH. ND = non-detected. The position of the primers used for amplifying the L region is depicted in (**A**). (**C–F**) The stimulatory potential of the indicated doses of input (**C** and **E**), LGP2 IP or ctrl IP RNA (**D** and **F**) was assessed by IFN-β promoter luciferase reporter assay in HEK293 cells. (**B–F**) One experiment of two is shown.

The following figure supplements are available for figure 5:

**Figure supplement 1**. 200 ng of RNA isolated from HeLa cells infected with EMCV WT, Zn$_{C19AC22A}$ or ΔL at MOI 1 for 16 hr (HeLa EMCV RNA) was transfected into MDA5-sufficient or MDA5-deficient bone marrow-derived DCs in presence of ribavirin.

These data suggest that the EMCV L AS RNA sequence rather than the position of the L region in the EMCV genome is the key determinant in innate stimulation. To further address this issue, we introduced the EMCV L antisense RNA sequence into the influenza virus NS segment, a known RIG-I agonist

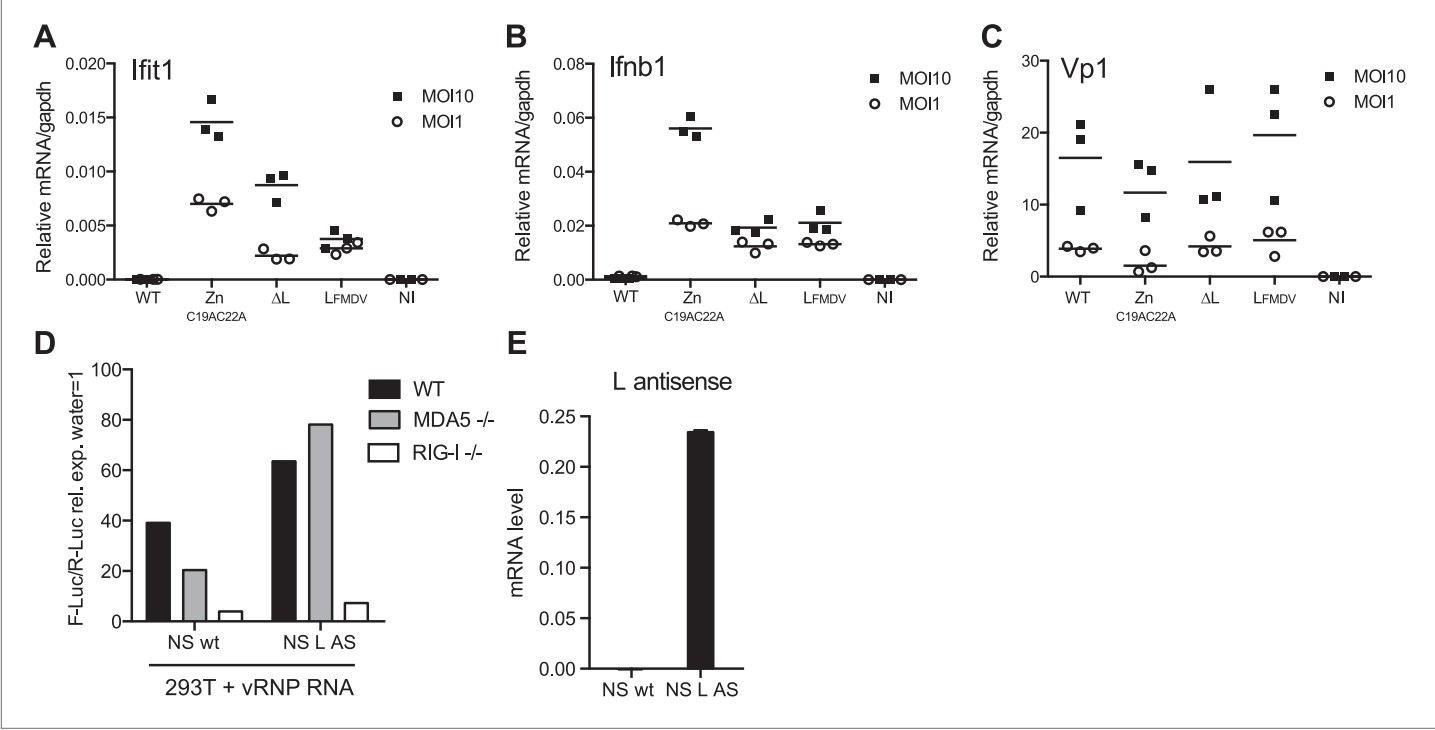

**Figure 6**. L region RNA is required for IFN-α/β production in response to EMCV. (**A–C**) IFNAR1-deficient GM-CSF bone marrow-derived DCs were either not infected (NI) or infected with the indicated viruses at an MOI of 1 or 10. Levels of Ifit1 (**A**), Ifnb1 (**B**) or EMCV Vp1 (**C**) RNA were analysed 16 hr later by quantitative RT-PCR and normalised to gapdh. (**D** and **E**) 293T cells were transfected with the influenza vRNP reconstitution system using the influenza NS wt segment or NS segment carrying the L AS RNA sequence and incubated for 24 hr. RNA was then extracted and the IFN stimulatory activity was tested on MEF wt, MDA5 or RIG-I deficient by luciferase assay (**D**) and the expression of the L AS sequence was verified by strand-specific RT-qPCR with primer specific for the L antisense RNA (**E**). (**A–E**) One experiment of two is shown.

(*Rehwinkel et al., 2010*). However, as shown in *Figure 6D,E*, the resulting chimeric NS RNA remained dependent on RIG-I and not MDA5 for its stimulatory activity. This suggests that the L region antisense sequence needs to be somehow processed or released in the context of EMCV infection to trigger an MDA5-dependent response and this does not happen in the context of the influenza virus NS segment.

Finally, we assessed whether L region RNA is also important for responses to EMCV in vivo. Mice deficient in IFNAR1 (to allow replication of the attenuated viruses) were infected with EMCV WT, EMCVΔL and EMCV Zn$_{C19AC22A}$ strains and the outputs of innate immune stimulation were measured. Higher levels of IFN-α were found in the serum of mice 24 hr after infection with EMCV Zn$_{C19AC22A}$ when compared to EMCVΔL and WT (*Figure 7A*). In addition, the expression of *Ifit1* and *Ifnb1* was also much higher in hearts from mice infected with EMCV Zn$_{C19AC22A}$ compared to mice infected with EMCVΔL and WT (*Figure 7B,C*). Measurement of VP1 mRNA in hearts confirmed broadly similar levels of infection by all viruses (*Figure 7D*). We conclude that the presence of L region RNA independently of the function of the L protein is important for innate immune responses to EMCV infection in vivo.

## Discussion

Several types of RNA can trigger MDA5-dependent responses in experimental settings (*Malathi et al., 2007*, *2010*; *Pichlmair et al., 2009*; *Luthra et al., 2011*; *Züst et al., 2011*; *Feng et al., 2012*; *Triantafilou et al., 2012*) but it remains unclear which RNAs serve as natural MDA5 agonists during virus infection. For example, some of the RNA species extracted from infected cells may not have access to MDA5 during infection or may be produced in quantities or at times that are irrelevant for innate immune recognition. One way to identify those RNAs most likely to be relevant agonists in infected cells is to isolate them directly from RLR complexes present in those cells (*Baum and García-Sastre, 2010*; *Rehwinkel et al., 2010*). In this study, we describe a method to isolate relevant MDA5 agonists from cells infected with EMCV by immunoprecipitation of LGP2/RNA complexes. We show that this

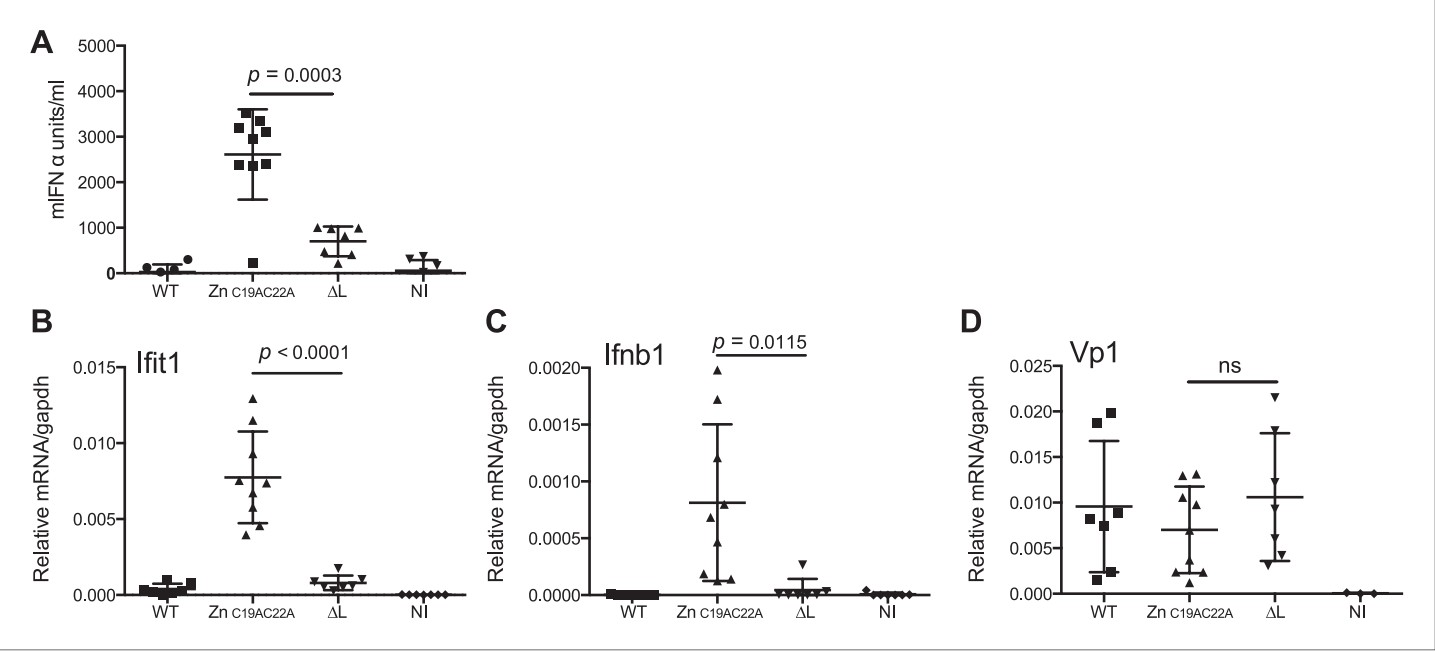

**Figure 7**. The L region is required for IFN-α/β responses to EMCV in vivo. (**A** and **B**) IFNAR1 KO mice were injected intraperitoneally with PBS (not infected; NI) or with 10ᴱ6 pfu of the indicated viruses. (**A**) mIFN-α levels in the serum were measured by ELISA after 24 hr. (**B–D**) RNA was extracted from hearts 24 hr postinfection and analysed by quantitative RT-PCR for levels of Ifit1 (**B**), Ifnb1 (**C**), or EMCV Vp1 (**D**) RNA normalised to gapdh expression. Combined results of two independent experiments are shown. Each symbol represents an individual mouse. ns = not significant.

method simultaneously enriches for MDA5 stimulatory activity and for a discrete species of antisense RNA mapping to the EMCV L region. Notably, we establish a causal connection between the two events by showing that deletion of the L region from the virus genome abrogates the association of stimulatory RNA with LGP2 in infected cells and reduces the innate stimulatory capacity of the virus, both in vitro and in vivo. Finally, we show that in vitro synthesis of L region antisense RNA is sufficient to generate an MDA5 agonist. Our results demonstrate that the antisense L region of EMCV associates with LGP2 and is a key determinant of MDA5 stimulation. Curiously, this determinant is derived from the same region that encodes a protein product that suppresses IFN induction. The fact that deletion of this region in EMCV reduces viral stimulatory activity is therefore counterintuitive and strengthens the notion that it is a physiologically-relevant major determinant of innate immunity to the virus.

The finding that pulldown of LGP2 allows isolation of MDA5 stimulatory RNA is in agreement with the fact that both RLRs are required for IFN production upon EMCV infection (*Figure 1—figure supplement 1*) and with previous reports describing LGP2 as a positive regulator of MDA5 signalling (*Venkataraman et al., 2007*; *Satoh et al., 2010*; *Bruns et al., 2012*; *Childs et al., 2013*). Interestingly, despite numerous attempts and experimental permutations, we have never been able to isolate stimulatory RNA directly from MDA5 pulldowns (data not shown). One explanation may be that LGP2 possesses higher avidity for RNA than MDA5 (*Bruns et al., 2012*; *Childs et al., 2013*) and is therefore more prone to retain bound agonist during cell lysis and precipitation. Another possibility relates to the fact that MDA5 forms filaments along RNA upon activation (*Peisley et al., 2011, 2012*; *Berke and Modis, 2012*; *Berke et al., 2012*; *Wu et al., 2012*) and that such structures—when formed in infected cells—may be difficult to isolate. These considerations raise the question of whether LGP2 precipitation selects relevant MDA5 agonists. For example, given that LGP2 can bind to stimulatory RNAs in vitro (*Figure 2F*), is it possible that the complexes that we precipitate result from post-lysis 'mopping-up' of stimulatory RNA that would never in the intact cell have come into contact with MDA5? We believe this is unlikely based on the fact that the functional relevance of the L region identified in the LGP2 pulldowns could be validated in the context of infection using mutant viruses. Furthermore, we find that LGP2 pulldown both enriches for MDA5 agonists and depletes them from input material. Therefore, it appears that the universe of LGP2-bound RNAs largely overlaps with that of MDA5 agonists in EMCV-infected cells. Although we do not exclude the possibility that there are MDA5 agonists that

are not captured by LGP2 pulldown (see below), our findings suggest a model where LGP2 pre-selects RNAs for MDA5 activation prior to the formation of MDA5 filaments. Whether the two proteins form a physical complex remains to be established—it is possible that LGP2 acts upstream of MDA5 to 'handover' agonistic RNA and that interaction with MDA5, if any, may be indirect or transient.

Deep sequencing of LGP2-associated stimulatory RNA was revealing. We were not able to identify specific enrichment for any human sequences (data not shown), a finding that does not support a dominant role for RNAseL cleaved self RNA in MDA5 activation (*Malathi et al., 2007*). This is in agreement with another recent publication that also failed to detect a significant role for RNaseL activity in EMCV-dependent IFN induction (*Feng et al., 2012*). Instead, the sequencing data, confirmed by strand-specific RT-PCR analysis, identified a different species of small RNA that was highly enriched in LGP2 pulldowns. This is a fragment of 171 nucleotides that corresponds to the antisense RNA in the L region of the genome. Its short length and failure to be accompanied by the complementary strand are at odds with the notion that MDA5 is activated by long dsRNA (*Kato et al., 2006*, *2008*). However, recent data indicate that MDA5 can form filaments along short dsRNA of around 100 nucleotides in length (*Peisley et al., 2011*). Moreover, the fact that LGP2-associated RNA appears to be single-stranded is in agreement with the observation that the stimulatory activity of the LGP2-associated RNA is sensitive to RNases that degrade ssRNA (*Figure 2E*). Interestingly, this LGP2-associated RNA is also sensitive to dsRNase treatment, which suggests the presence of base-paired regions important for MDA5/LGP2 recognition. Consistent with that notion, heat denaturation destroyed its activity (data not shown). In silico analysis reveals the potential of the L antisense RNA to form hairpins (data not shown), and such secondary and tertiary structure features may help to efficiently trigger MDA5 activation. Interestingly, by in vitro transcription of a sequence corresponding to the L antisense region we have succeeded in creating de novo an RNA that triggers MDA5 activation (*Figure 4*). What endows this RNA with MDA5 agonistic activity and whether the in vitro transcribed version fully corresponds to that naturally formed in EMCV-infected cells will require further investigation.

Another intriguing question is how a fragment of L AS RNA is generated upon viral infection. We have not been able to find a report describing the generation of small antisense RNA fragments during EMCV replication. However, Northern blots of RNA from EMCV-infected cells screened with an L AS sequence probe reveal many fragments smaller than the EMCV genome ranging between 200 and 2000 nucleotides (data not shown). Moreover, a previous study has reported the generation of a short subgenomic RNA fragment derived from the degradation of the non-polyadenylated genome of flaviviruses by the exonuclease XRN1 (*Funk et al., 2010*). It is possible that a similar mechanism could generate an L antisense RNA fragment. Alternatively, stress granule formation, known to interfere with picornavirus replication (*Borghese and Michiels, 2011*), could potentially lead to degradation of viral RNA and generate the L region antisense RNA agonist. Interestingly, MDA5 and LGP2 have been recently shown to associate with stress granules (*Onomoto et al., 2012*; *Langereis et al., 2013*).

Picornavirus replication involves a stage where the positive-stranded genome anneals with a newly-formed negative strand in a full-length double-stranded RNA structure known as the replicative form (RF). Negative-strand RNA from the RF then serves as the template to produce new progeny viral genomes through a replicative intermediate (RI) consisting of many positively-stranded progeny RNA genomes in the process of being synthesised and still partially hybridised to the full-length antisense RNA template. A recent study showed that negative-strand RNA synthesis is absolutely required for IFN induction during infection of mengovirus, a strain of EMCV (*Feng et al., 2012*). Our results that a small RNA derived from the negative-strand RNA of EMCV associates with LGP2 and activates MDA5 is in line with that report. However, it has also been shown that both the RF and RI forms can activate MDA5 upon transfection (*Feng et al., 2012*; *Triantafilou et al., 2012*). It is conceivable that RF and RI forms could be processed by RNases within the cell following transfection or infection, releasing the L antisense fragment that associates with LGP2. Alternatively, it is possible that the L region antisense fragment enriched in our IP is only one of several agonists generated upon EMCV replication and that the RF and RI forms constitute a different set of agonists that do not strongly associate with LGP2 and have therefore been missed in our approach. This hypothesis is consistent with our data, which show that EMCVΔL viruses still possess the ability to induce low levels of IFN and IFIT-1 (*Figures 6 and 7*) and generate stimulatory RNA (*Figure 5*). Such data suggest the existence of at least one additional stimulatory RNA species beside the L region antisense fragment and help explain why EMCVΔL is an attenuated strain: the additional stimulatory RNA means that it cannot fully evade innate immune detection while the lack of the L protein means that it cannot suppress the consequences of detection. Although it remains to be established

whether this putative additional stimulatory RNA species corresponds to the RF/RI (or even acts via MDA5—*Figure 5—figure supplement 1*), these observations suggest the need to determine to what extent different innate immune stimuli dominate at different stages of infection.

Our investigation has resulted in the identification of a specific region of the EMCV negative-strand RNA as a determinant of innate immunity to the virus. To our knowledge, this is the first study to identify an MDA5 agonist bound to LGP2 in infected cells. Picornaviruses constitute a large family of viruses that encompasses many human and animal pathogens including some of economical or medical importance including Poliovirus, Coxsackievirus, Rhinovirus, and EMCV (*Tuthill et al., 2010*). The L regions of picornaviruses can show wide variation and it will be interesting to determine to what extent L region RNA or other small RNAs can act as MDA5 agonists across picornavirus genera. Elucidating the molecular basis of picornavirus detection will help understand the general rules underlying innate virus detection and may suggest new strategies to control viral infection.

## Material and methods

### Reagents

The human LGP2 sequences were amplified from cDNA derived from IFN-A/D (PBL Assay Science, Piscataway, NJ)-treated HEK293 cells using a forward primer containing the 3xFLAG epitope sequence with the following oligos (FW LGP2: 5'gccgccatggactacaaagaccatgacggtgattataaagatcatgacatcgattacaag-gatgacgatgacaaggagcttcggtcctaccaatggga-3', RV LGP2: 5'-tcagtccagggagaggtccga-3'). PCR products were then introduced into the pcDNA3.1 TOPO plasmid (Life Technologies, Foster City, CA) to generate the 3xFLAG LGP2 expression constructs. For production of the recombinant LGP2 protein, the 3xFLAG-human LGP2 sequence was amplified using the forward primer 5'-actcgagttatggactacaaagaccatgacgg-3' and the reverse primer 5'-ttgcggccgctcagtccagggagaggtccga-3' and cloned into the XhoI and NotI sites of the pBacPAK-His3-GST plasmid. Recombinant 3xFLAG LGP2 was expressed as a GST-tagged protein in SF9 insect cells using a baculovirus expression system and purified on a single step by affinity chromatography using Glutathione Sepharose matrix (GE Healthcare Life Sciences, Buckinghamshire, UK). The 3xFLAG LGP2 was eluted by GST tag cleavage using in-house 3C enzymatic digestion. A final polishing step was then performed on a superdex 200 10/300 GL column (GE Healthcare). Protein purity was then verified on an acrylamide gel and the protein yield was quantified using a Nanodrop apparatus (ThermoScientific, Wilmington, DE).

The M2 (Sigma) and IgG1 isotype match control (BD Pharmingen, San Diego, CA) antibodies were used for immunoprecipitation. M2 antibody was used at 1/5000 dilution for Western blot. Anti-mouse HRP antibody (Southern Biotech, Birmingham, AL) was used at 1/10,000 dilution for Western blot. IFN A/D (PBL Assay Science) was used at 100 units/ml for 24 hr to pre-treat cells. Ribavirin powder (Sigma) was reconstituted in DMSO and used at 4 mM final concentration.

### Cells

HeLa, BHK21, and Vero cells were from Cancer Research UK (CRUK) Cell services. HEK293 selected for responsiveness to RLR agonists were previously described (*Pichlmair et al., 2009*). WT, $Ifih1^{-/-}$, $Ifih1^{+/-}$, $Ddx58^{-/-}$, $Ddx58^{+/+}$, $Ddx58^{-/+}$, $Dhx58^{-/-}$ and $Dhx58^{+/+}$, MEFs were generated as previously described (*Kato et al., 2006*), immortalised with simian virus 40 large T antigen and selected on puromycin (final concentration, 2 mg/ml) for 2 weeks. HeLa and BHK21 cells were grown in 10% FCS-containing minimum essential medium (CRUK) or Glasgow media (Life Technologies), respectively. All other cell lines were grown in Dulbecco's modified Eagle's medium containing 10% FCS and 2 mM glutamine. Mouse BM-DCs were generated using GM-CSF as described previously (*Inaba et al., 1992*). Briefly, femur and tibia were collected from both hindquarters. Bones were flushed with RPMI 1640 (Life Technologies) media containing 10% FCS, 100U/ml Penicillin/Streptomycin, 5 µM β-mercaptoethanol and 200 units/ml of GM-CSF (CRUK) and passed through a 70 µM cell strainer. Cells were then cultured for 5 days with medium renewal every 2 days.

### Viruses

Influenza A virus (IAV) (PR8 strain) and EMCV were as previously described (*Pichlmair et al. 2009*). EMCV $Zn_{c19AC22A}$, EMCV ΔL and EMCV $L_{FMDV}$ (mengovirus strain), were produced as described (*Feng et al., 2012*). Other EMCV mutants were generated as follows: pEC9 EMCV WT, pEC9 EMCV ΔL, pEC9 EMCV ΔL$_{ac}$ and pEC9 EMCV ΔL$_{zn}$ plasmids (kind gift from Ann C Palmenberg) were in vitro transcribed to generate full length EMCV RNA using T7 Megascript kit (Ambion) as per manufacturer's

instructions. Products were then digested with DNase I and purified with phenol:chloroform:isoamyla lcohol (25:24:1), followed by chloroform extraction and ethanol precipitation. RNAs were transfected into HeLa cells to generate viruses, which were subsequently amplified on BHK21 cell monolayers until cytopathic effects were observed. Cell lysates were then freeze-thawed three times, cleared and centrifuged for 2 hr at 22,000 rpm (SW 32 Ti Rotor, Beckman ultracentrifuge) at 4°C on 20% sucrose cushion to purify viral particles. The pelleted viral particles were resuspended in 10 mM Tris pH7, 2 mM $MgCl_2$ containing buffer and viral content quantified by plaque assay on Vero cells.

## Mouse infection

IFNAR1 KO mice were obtained from Michel Aguet (University of Lausanne) and backcrossed 14 times to C57BL/6J. The mice were bred at CRUK or at St Mary's Hospital (kind gift from Cecilia Johansson) in specific-pathogen free conditions. For infection, 10- to 12-week-old C57BL/6-IFNAR1 KO mice were injected intraperitoneally with $10^6$ pfu of the indicated virus in 200 µl of PBS. Control mice were injected with 200 µl of PBS. Serum and organs were collected from culled animals 24 hr after injection. All animal experiments were performed in accordance with national and institutional guidelines for animal care and were approved by the London Research Institute Animal Ethics Committee and by the UK Home Office (project licence PPL 80/2309).

## RNAs

The IVT RNA used as a positive control for IFN induction was the IVT neomycin sequence previously described (*Rehwinkel et al., 2010*). The in vitro transcribed RNA derived from the L AS and sense RNA sequence were generated by in vitro transcription using the T3 and Sp6 Megascript kit (Ambion Life Technologies) using the PCR with primer fw 5′-gcgcactctctcacttttga-3′ and rv 5′-aaatttaggtgacac-tatagaagcgctcgaaaacgacttccatgt-3′ or fw 5′-aaaattaaccctcactaaagggagaacttgcgcgcactctctcac-3′ and rv 5′-tcgaaaacgacttccatgtct-3′ for the production of the L AS and sense templates.

Viral genomic RNAs were extracted from purified viral suspension using Trizol LS reagent (Life Technologies) as per manufacturer's recommendations. For RNA extracted from infected HeLa cells, HeLa cells were infected with EMCV or IAV at an MOI of 1 for 16 hr before total RNA was extracted using Trizol. For separation of RNA into ss and dsRNA fractions, ssRNAs were first precipitated in presence of 2M LiCl. Double-stranded RNAs were then ethanol precipitated from the supernatant in presence of 0.7M LiCl. RNA pellets were washed with 70% ethanol, dried, and resuspended in RNase free water.

Calf intestinal phosphatase (New England Biolabs, Ipswich, MA), Terminator (Epicenter Biotechnologies, Madison, WI), RNaseT1 (Ambion Life Technologies), Polyphosphatase (Epicenter Biotechnologies), RNase A (Sigma, St. Louis, MO) and RNase III (Ambion Life Technologies) were used as per manufacturer's recommendations. A control reaction omitting the enzyme was carried out in parallel. RNA was recovered by extraction with phenol:chloroform:isoamylalcohol (25:24:1), followed by chloroform extraction and precipitation with ethanol and sodium acetate in the presence of glycogen (Ambion). All RNAs were quantified using a Nanodrop apparatus.

For quantitative RT-PCR analysis of *Ifnb1*, *Ifit1*, *IFNB1*, *gapdh*, *GAPDH*, *VP1* and *L* genes, RNA was extracted from $1 \times 10^5$ MEFs, BM-DCs or HeLa cells using the RNeasy kit (Qiagen, Valencia, CA) or from infected mouse organs using Trizol.

## Construction of the NS L AS influenza segment and vRNP reconstitution assay

To generate the NS L AS Influenza segment, we amplified the L EMCV fragment amplified using the forward primer 5′-aaaccatggatggccacaaccatggaac-3′ and the reverse primer 5′-aaaccatggctgtaactc-gaaaacgactt-3′ and introduced it into NcoI site of the pPolI_NS plasmid encoding the NS segment sequence of Influenza WSN strain (gift from Ervin Fodor). vRNP were then reconstituted in HEK 293T cells as previously described in *Rehwinkel et al. (2010)*.

## Quantitative RT-PCR analysis

RNA was treated with DNase I (Qiagen) prior to reverse transcription using superscript II reverse transcription reagents (Life Technologies) according to the manufacturer's instructions. PCR was performed with TaqMan Universal PCR master Mix (Applied Biosystem Life Technologies) and the following taqman reagent assays: mouse Ifit1 ID Mm00515153_m1, mouse Ifnb1 ID Mm00439546_s1, mouse gapdh ID 4308313, human IFNB1 ID Hs02621180_s1, human gapdh ID 402869. To detect viral RNA, Express Syber GreenER reagent (Life Technologies) was used in combination with the following

primers: fw 5'-gcgcactctctcacttttga-3' and rv 5'-tcgaaaacgacttccatgtct-3' for detection of the L region or fw 5'-cctcttctcccccctttgtgt-3' and rv 5'-caggtccggcactataaacc-3' for detection of the VP1 region. Data were normalized to levels of gapdh.

For strand-specific detection of viral RNA, RNA was first reverse transcribed using Superscript II (Life Technologies) in the presence of primers specific for L antisense (5'- ggccgtcatggtggcgaataagcg-cactctctcacttttga-3') or VP1 antisense (5'- ggccgtcatggtggcgaataacaggtccggcactataaacc-3'). cDNAs were then subjected to Exonuclease I (New England Biolabs) treatment and purified using Qiaquick PCR purification kit (Qiagen). In the next step, quantitative RT-PCR was performed in presence of Express Syber GreenER (Invitrogen) using the following primers: common forward primer 5'-aataaatcataaggccgt-catggtggcgaataa-3' in combination with either L reverse primer 5'-aataaatcataatcgaaaacgacttccatgtct-3' or 1D reverse primer 5'- aataaatcataacctcttctcccccctttgtgt-3' to detect L or 1D region respectively. RNA levels were calculated by comparison with a dilution curve of cDNA from EMCV-infected cells.

## Detection of IFN stimulatory activity

IFN-β luciferase reporter assay: $2.5 \times 10^5$ of HEK293 cells or IFN-A/D treated MEFs were transfected with 200 ng of p125Luc (gift from T Fujita, Kyoto university, Japan) and 50 ng pRL-TK (Promega, Madison, WI) using Lipofectamine 2000 (Life Technologies) as per manufacturer's instructions. Cells were incubated for 6–8 hr and were then transfected with water (mock control) or with test or control RNAs using lipofectamine. Luciferase activity was analysed in cell lysates 12 to 16 hr later using the Dual luciferase reporter Assay system (Promega). In all cases, firefly luciferase values were divided by Renilla luciferase values to normalise for transfection efficiency. All data are shown as fold increase relative to reporter cells transfected with water alone.

MEF assay: MEFs were pre-treated with 100 units/ml of IFN-A/D (PBL interferon) for 24 hr and plated into 24-well plates at $0.5 \times 10^5$ cell/well. The cells were then transfected with test or control RNAs using lipofectamine. After overnight culture, mIFN-α was measured in cell supernatants by ELISA as described previously (*Rehwinkel et al., 2010*) or MEF RNA was extracted with an RNeasy kit (Qiagen) following manufacturer's instructions for RT-PCR analysis of Ifit1 and Ifnb1 induction.

HeLa assay: HeLa cells were pre-treated with 100 units/ml of IFN-A/D (PBL Assay Science) for 24 hr and plated into 24-well plates at $0.5 \times 10^5$ cell/well. The cells were then transfected with the indicated amounts of test or control RNAs using lipofectamine. After overnight culture, HeLa RNA was extracted for RT-PCR analysis of IFNB1 induction.

## LGP2 immunoprecipitation

Around 10–15 million HeLa cells were transfected with 30 μg of 3xFLAG LGP2 IP plasmids using lipofectamine reagent following the manufacturer instructions. The cells were incubated for 16 hr, lipofectamine was washed away and the cells were incubated in fresh medium for 6–8 hr. HeLa cells expressing the 3xFLAG constructs were then infected with EMCV at an MOI of 1 for 16 hr. The cells were subsequently washed and lysed in lysis buffer (10 mM Tris pH 7.4, 2.5 mM MgCl$_2$, 200 mM NaCl, 0.5% NP40, 1X protease inhibitor cocktail [Roche, Mannheim, Germany], 0.5 U/ml RNasin [Promega]). Lysate was incubated for 30 min on ice, cleared by centrifugation for 10 min and the supernatant was collected. At this stage a small fraction of the input was collected for protein and RNA extraction. 5 μg of M2-anti-FLAG (Sigma) or IgG1 isotype control antibody was added to 500 μl of lysate and incubated on a rotating shaker for 1 hr at 4°C. 300 μl of washed Gamma Bind Plus Sepharose Beads (GE Healthcare Bioscience AB) were then added to the mixture for another 1 hr. The beads were then precipitated by centrifugation and washed four times with 1 ml of lysis buffer. The beads were then split into two samples for protein or RNA extraction. Proteins were extracted from Protein/RNA complexes by boiling the beads for 5 min in SDS sample buffer for Western blot analysis. RNAs were purified from the beads by Phenol/chlorophorm extraction followed by ethanol precipitation. RNA was then quantified using a Nanodrop apparatus and the same amount of RNA from each sample was tested for IFN stimulatory activity as indicated.

For the IP using recombinant LGP2, 5 μg of the purified protein was incubated with 20 μg of total EMCV-infected RNA (MOI 1; 16 hr) in lysis buffer for 1 hr at 4°C on a rotating shaker. The rest of the IP was performed the same way as for the LGP2 IP from infected cells.

## Deep sequencing of RNA

Input material pooled from five replicate cultures was subjected to immunoprecipitation with anti-FLAG or control IgG1 antibody (see above). RNA extracted from 10 independent IPs was pooled

and ribosomal RNA was removed using Ribo-Zero rRNA Removal Kit (Human.Mouse.Rat) (Epicenter RZH1046, Madison, WI) according to the manufacturer's protocol. After ribosomal RNA removal, the RNA was tested for its ability to activate the IFN-β promoter luciferase reporter. The RNA was then prepared for Illumina sequencing using an optimised version of the original Directional mRNA-Seq Library Prep protocol (Pre-Release Protocol Rev.A). The original protocol was optimised from 1 μg of total RNA input to a total input of up to 173 ng. Reagents supplied by Illumina included 10× v1.5 sRNA 3′ Adaptor; SRA 5′ Adaptor; SRA RT primer; PCR Primer GX1/GX2. The remaining reagents recommended on the protocol were outsourced from alternative vendors. To analyse both coding and non-coding RNA regions, the poly(A) purification step was omitted. After fragmentation of the total RNA (optimised incubation of 2 min at 80°C), the RNA was checked for efficient fragmentation sizing, ribosomal RNA contamination and RIN using the Agilent 2100 Bioanalyzer QC Pico RNA chip. After library preparation, the PCR reaction was optimised by replacing the recommended PCR polymerase with Kapa Hifi DNA Polymerase (KAPA Biosystems, Wilmington, MA) in addition to optimising the number of PCR cycles from 12 to 18 cycles using the recommended PCR cycling conditions. The final library preparation was then size selected at 150 bp–450 bp using a 2% Agarose E-gel (Invitrogen) to remove unwanted regions in excess of 500 bp which was detected using the Agilent 2100 Bioanalyzer QC DNA 1000 chip. Next generation sequencing and library preparation was performed in the Advanced Sequencing Facility (ASF) at the London Research Institute on the Genome Analyzer IIx (GAIIx) with a single-end 72 bp sequencing run alongside a PhiX control of 1–5% in every lane of the flowcell. Sequencing typically generated ~30 million 60 bp single-end reads.

Alignment to the EMCV strain pEC9 genome (genbank accession ID DQ288856) was performed using Bowtie (*Langmead et al., 2009*; version 0.12.7) permitting a maximum of three mismatches per read. Genome-wide coverage was calculated using the genomeCoverageBed function in BEDTools (*Quinlan and Hall, 2010*; version 2.16.2) and all subsequent plots were generated using the statistical programming language, R (R core team 2012, R: A language and environment for statistical computing. R Foundation for Statistical Computing, Vienna, Austria; version 2.15). To generate more comparable data across samples the number of reads were normalised to the number of reads per million.

## Statistical analysis

An unpaired two-tailed Student's *t* test was used to determine statistically significant differences. p values of less than 0.05 were considered statistically significant. GraphPad Prism version 6 for Macintosh (GraphPad Software) was used for statistical analysis of data.

## Acknowledgements

We thank Ann C Palmenberg, Takashi Fujita, Ervin Fodor, Shizuo Akira and Cecilia Johansson for generous gifts of reagents and mice. We thank Cecilia Johansson, Barbara U Schraml and Kathryn Snelgrove for reading the manuscript, Mike Skinner for advice and all members of the Immunobiology laboratory for helpful discussions and comments. SD is a recipient of Cancer Research UK and Marie-Curie long-term fellowships (FP7-PEOPLE-2010-IEF-273483). CRS is funded by Cancer Research UK, a prize from Fondation Bettencourt-Schueller, and a grant from the European Research Council (ERC Advanced Researcher Grant AdG-2010-268670). QF is supported by Mosaic grant (NWO-017.006.043) and FJM by Echo grant (NWO-CW-700.59.007) from the Netherlands Organisation for Scientific Research (NWO).

## Additional information

### Funding

| Funder | Grant reference number | Author |
|---|---|---|
| Cancer Research UK | N/A | Safia Deddouche, Delphine Goubau, Jan Rehwinkel, Probir Chakravarty, Sharmin Begum, Pierre V Maillard, Annabel Borg, Nik Matthews, Caetano Reis e Sousa |

| Funder | Grant reference number | Author |
|---|---|---|
| Medical Research Council | | Jan Rehwinkel |
| Marie Curie | | Safia Deddouche |
| Fondation Bettencourt-Schueller | N/A | Caetano Reis e Sousa |
| European Research Council | AdG-2010-268670 | Caetano Reis e Sousa |
| Netherlands Organisation for Scientific Research - Mosaic grant | NWO-017.006.043 | Qian Feng |
| Netherlands Organisation for Scientifique Research - ECHO Grant | NWO-CW-700.59.007 | Frank J M van Kuppeveld |

The funders had no role in study design, data collection and interpretation, or the decision to submit the work for publication.

## Author contributions

SD, Conception and design, Acquisition of data, Analysis and interpretation of data, Drafting or revising the article; DG, Acquisition of data, Analysis and interpretation of data, Drafting or revising the article; JR, Contributed unpublished essential data or reagents; PC, Analysis and interpretation of data; SB, NM, Acquisition of data; PVM, QF, FJMK, Analysis and interpretation of data, Contributed unpublished essential data or reagents; AB, Provided reagents, Conception and design; CRS, Conception and design, Analysis and interpretation of data, Drafting or revising the article

## Ethics

Animal experimentation: All animal experiments were performed in accordance with national and institutional guidelines for animal care and were approved by the London Research Institute Animal Ethics Committee and by the UK Home Office (project licence PPL 80/2309).

## Additional files

### Major dataset

The following previously published dataset was used:

| Author(s) | Year | Dataset title | Dataset ID and/or URL | Database, license, and accessibility information |
|---|---|---|---|---|
| Martin LR, Neal ZC, McBride MS, Palmenberg AC, Aminev A, Hill M, Groppo R | 2000, 2005 | Encephalomyocas virus strain pEC9, complete genome | DQ288856.1; http://www.ncbi.nlm.nih.gov/nuccore/DQ288856.1 | Publicly available at NCBI GenBank (http://www.ncbi.nlm.nih.gov/genbank/). |

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
