## [Decision Letter]

Thank you for sending your work entitled “Identification of a natural MDA5 agonist in picornavirus infected cells” for consideration at *eLife*. Your article has been favorably evaluated by a Senior editor and 3 reviewers, one of whom is a member of our Board of Reviewing Editors.

The Reviewing editor and the other reviewers discussed their comments before we reached this decision, and the Reviewing editor has assembled the following comments to help you prepare a revised submission.

In the manuscript “Identification of a natural MDA5 agonist in picornavirus infected cells”, Deddouche et al. investigate the properties of MDA5 stimulatory RNA during EMCV infection. The authors use total RNA of EMCV-infected HeLa cells as a source for natural MDA5 ligands, from which they purify RNA based on affinity for the RIG-I like receptor LGP2. The authors find that RNA bound to LGP2 is enriched for MDA5-dependent stimulatory activity. Using next-generation sequencing, the authors observe an enrichment of viral sequences in the precipitated RNA, and in particular of antisense-transcripts from the EMCV L-region. By employing deletion mutants for this region, as well as point mutants, they attribute the majority of MDA5-activity generated in EMCV-infected cells to the L-region antisense RNA. Of note, this feature is clearly independent of the L-Protein encoded therein.

The finding that a specific region transcribed from the EMCV genome functions as a natural and preferential ligand of LGP2/MDA5 is novel and significant. This represents an important step towards defining the ligand of LGP2 and MDA5. In addition to its conceptual novelty, the study is important because of many infectious diseases caused by positive strand RNA viruses. However, there are several major issues that should be addressed before the paper can be considered for publication.

1) As it stands now, the definition of this ‘L’ RNA ligand is still ‘fuzzy’. In particular, the authors stated that the L fragments generated by in vitro transcription or chemical synthesis did not stimulate the MDA5/LGP2 pathway. Thus, it is not clear what makes the L antisense RNA fragment a unique ligand for LGP2. At the minimum, the authors should perform deletion and/or point mutagenesis of the L RNA sequence to determine if there are specific sequence(s) in this region that renders it a good LGP2 ligand. The EMCV mutant Zn_C19AC22A_ can be a good starting point for mutagenesis because this virus can induce IFN-β quite well. It is formally possible that any RNA sequence, when placed in this region, could be a ligand for LGP2. It is important to have a better characterization of the antisense L RNA fragment, especially given that this RNA could not be detected by Northern blotting and that currently there is no explanation for how this RNA is generated in the EMCV infected cells.

2) Of note, the study does not identify the EMCV RNA structure that activates MDA5 but the sequence that preferentially binds to LGP2. The authors should therefore amend the claim that it identities a natural MDA5 agonist, or that the L-region RNA itself constitutes an MDA5 agonist. Given that L-region RNA on its own fails to activate MDA5, one must assume that the antisense L-region RNA enriched by LGP2 is merely contained in the same RNA strand as the actual MDA5 agonist, which also remains unidentified. Does introduction of the antisense EMCV L-region into previously inactive RNA transcripts render them MDA5 agonists?

3) How long is the antisense L RNA that is proposed to be the LGP2 ligand? Is it a uniform size or heterogeneous? Is it possible that the RNA ligand is actually longer than those detected by RNA-seq because the parts of RNA not bound (protected) by LGP2 are degraded by RNases? This could be one of the reasons that the in vitro transcribed antisense L RNA has no activity.

4) While a lot of evidence is presented that suggests that LGP2 delivers the agonist to MDA5, no direct evidence is presented. Especially considering the fact that pure L region antisense fragment does not recapitulate the observed phenotype, more evidence is required to corroborate the claim that LGP2 serves as a means to preselect agonists for MDA5 activation in the context of EMCV infection. Is MDA5 associated with the RNA in the LGP2 pulldowns? Do MDA5 and LGP2 interact after ligand capture? Does this interaction induce the formation of MDA5 filaments or IFN expression?

5) Figure 5: There is still a substantial level of IFN-β induction by the EMCV virus lacking the L RNA. Is this L-independent activity dependent on RIG-I or MDA5? If it is MDA5 or LGP2 dependent, it suggests that there are other MDA5 ligands in the virus infected cells. These could conceivably be dsRNA as previously thought as MDA5 ligands. Indeed, the authors showed that dsRNA isolated from EMCV infected cells induced IFN-β (Figure 2).

6) In cells infected by EMCV virus lacking the L region, can the dsRNA and ssRNA be isolated from the gels (Figure 2) and shown to stimulate IFN-β? This could help clarify where the antisense L RNA comes from and whether it has an indirect effect on the generation of dsRNA.

---

## [Author Response]

*1) As it stands now, the definition of this ‘L’ RNA ligand is still ‘fuzzy’. In particular, the authors stated that the L fragments generated by in vitro transcription or chemical synthesis did not stimulate the MDA5/LGP2 pathway. Thus, it is not clear what makes the L antisense RNA fragment a unique ligand for LGP2. At the minimum, the authors should perform deletion and/or point mutagenesis of the L RNA sequence to determine if there are specific sequence(s) in this region that renders it a good LGP2 ligand. The EMCV mutant Zn*_*C19AC22A*_
*can be a good starting point for mutagenesis because this virus can induce IFN-β quite well. It is formally possible that any RNA sequence, when placed in this region, could be a ligand for LGP2. It is important to have a better characterization of the antisense L RNA fragment, especially given that this RNA could not be detected by Northern blotting and that currently there is no explanation for how this RNA is generated in the EMCV infected cells*.

We have now been able to generate an L region antisense RNA by in vitro transcription that possesses MDA5-stimulatory activity (new Figure 4). In contrast, an RNA sequence derived from the sense strand does not possess activity (new Figure 4). As such, we can now state with more confidence that we have uncovered an MDA5 agonist in the form of RNA corresponding to the EMCV L antisense region. It is unclear why we had not succeeded in generating this agonist in our earlier experiments as the only “trick” that led to success was to use a primer that makes the transcript 4 nucleotides shorter than the one we had tried before. This means that there is additional work to be done in characterising the RNA in question and in determining the “rules” for MDA5 triggering. Such detailed characterisation will be the subject of future experiments.

In relation to the other comments by the reviewers, we agree that the mutagenesis study of Zn_C19AC22A_ EMCV would provide useful information. However, this is not a trivial exercise as it involves the rescue of hundreds of viruses. As it is, we have found it quite problematic to rescue and grow the mutant virus, which grows poorly. Nevertheless, we do show that 3 different EMCV mutants that carry either full or partial deletion of the L acidic region or the L zinc domain all display impaired ability of the viral RNA to interact with LGP2 (Figure 5).

At the suggestion of the reviewers, we also tested if any RNA sequence when placed in the L region can trigger a strong interferon response. For this, we replaced the L region of EMCV with that of another picornavirus (foot and mouth disease virus).

Notably, this recombinant EMCV behaves much like the EMCVΔL virus (amended Figure 6 of the paper), which indicates that the specific L region sequence of EMCV and not just the location within the genome is important for IFN production. We thank the reviewers for encouraging us to do this interesting experiment.

How the L antisense fragment is generated remains mysterious. We make this point clearer in the revised discussion by pointing out that nuclease digestion may be involved and that we are able to detect RNA smaller than the EMCV RNA full length ranging from 2000 to 200nts that light up with the L antisense probe (data not shown).

*2) Of note, the study does not identify the EMCV RNA structure that activates MDA5 but the sequence that preferentially binds to LGP2. The authors should therefore amend the claim that it identities a natural MDA5 agonist, or that the L-region RNA itself constitutes an MDA5 agonist. Given that L-region RNA on its own fails to activate MDA5, one must assume that the antisense L-region RNA enriched by LGP2 is merely contained in the same RNA strand as the actual MDA5 agonist, which also remains unidentified. Does introduction of the antisense EMCV L-region into previously inactive RNA transcripts render them MDA5 agonists*?

We thank the reviewers for these thoughtful comments. We feel that the claim that we have identified an MDA5 agonist is now warranted by the new data in Figure 4 showing that we can synthesise an MDA5 stimulatory RNA in vitro. But, as we do not show evidence of direct binding between MDA5 and the L antisense RNA, we have amended the title of the manuscript.

As suggested by the reviewers, we also have tried to introduce the EMCV L antisense sequence into a heterologous RNA. For that experiment, we chose to introduce it into the influenza virus NS segment, a known RIG-I agonist. As shown in the new Figure 6, however, the resulting RNA remains firmly dependent on RIG-I and not MDA5 for its stimulatory activity. This may mean that the sequence needs to be somehow processed or released to trigger an MDA5 dependent response and this does not happen in the context of the flu NS segment.

*3) How long is the antisense L RNA that is proposed to be the LGP2 ligand? Is it a uniform size or heterogeneous? Is it possible that the RNA ligand is actually longer than those detected by RNA-seq because the parts of RNA not bound (protected) by LGP2 are degraded by RNases? This could be one of the reasons that the in vitro transcribed antisense L RNA has no activity*.

We can indeed not exclude that partial degradation of the RNA does take place and that LGP2 protects a fragment of 171 nucleotides as revealed by our sequencing data. However, the fact that we can now synthesise this fragment and reveal stimulatory activity confirms that it is sufficiently long to activate MDA5. It is remains possible that the LGP2 agonist is actually smaller than 171nts. The full characterisation of this RNA will require further work, which we feel is beyond the scope of the present report.

*4) While a lot of evidence is presented that suggests that LGP2 delivers the agonist to MDA5, no direct evidence is presented. Especially considering the fact that pure L region antisense fragment does not recapitulate the observed phenotype, more evidence is required to corroborate the claim that LGP2 serves as a means to preselect agonists for MDA5 activation in the context of EMCV infection. Is MDA5 associated with the RNA in the LGP2 pulldowns? Do MDA5 and LGP2 interact after ligand capture? Does this interaction induce the formation of MDA5 filaments or IFN expression*?

To address the reviewers’ comments, we have tried to co-immunoprecipitate MDA5 and LGP2. To do this, we have coexpressed FLAG-tagged LGP2 together with HA-tagged MDA5 in HeLa cells but, oddly, we lose MDA5 detection when the cells are infected with EMCV (data not shown). We do not know the reason for this phenomenon but attribute it to the capacity of MDA5 to form insoluble filaments upon activation that are lost from the lysate during preclearing. We have attempted to circumvent the issue by using lysate from uninfected cells that has been spiked with RNA from EMCV infected Vero cells prior to MDA5 or LGP2 immunoprecipitation but this has failed to reveal any co-precipitation (data not shown). Because these experiments are negative they remain inconclusive and, therefore, we have not included them in the paper. Thus, we do not at present have proof that LGP2/MDA5/RNA form a tripartite complex and, consequently, have amended the term “complex” throughout the text. It is in fact possible that LGP2 acts upstream of MDA5 to remodel agonistic RNA or that its interaction with MDA5 is very transient and unlikely to be captured by biochemical means. We now refer to these possibilities in the revised Discussion. We thank the reviewers for allowing us to clarify the issue.

*5)*
Figure 5*: There is still a substantial level of IFN-β induction by the EMCV virus lacking the L RNA. Is this L-independent activity dependent on RIG-I or MDA5? If it is MDA5 or LGP2 dependent, it suggests that there are other MDA5 ligands in the virus infected cells. These could conceivably be dsRNA as previously thought as MDA5 ligands. Indeed, the authors showed that dsRNA isolated from EMCV infected cells induced IFN-β (*Figure 2*)*.

The notion that there are additional MDA5 triggers besides the L region antisense RNA is a possibility that we had considered in the original Discussion. We make this point clearer with textual revisions. As regards the question of whether the activity remaining in the EMCV delta L virus is MDA5 dependent, we now state in the revised Discussion that this point is unclear and have added a new figure to support that statement (new Figure 5—figure supplement 1).

*6) In cells infected by EMCV virus lacking the L region, can the dsRNA and ssRNA be isolated from the gels (*Figure 2*) and shown to stimulate IFN-β? This could help clarify where the antisense L RNA comes from and whether it has an indirect effect on the generation of dsRNA*.

We have isolated the dsRNA and ssRNA fractions from cells infected with EMCV virus lacking the L region as requested (data not shown). We have refrained from including the results in the revised manuscript as we are not entirely sure what they mean, especially in the face of the new data on generation of the MDA5 stimulatory activity in vitro.